# Thermodynamic Correlation between Liquid–Liquid Phase Separation and Crystalline Solubility of Drug-Like Molecules

**DOI:** 10.3390/pharmaceutics14122560

**Published:** 2022-11-22

**Authors:** Taiga Uekusa, Tomohiro Watanabe, Daiju Watanabe, Kiyohiko Sugano

**Affiliations:** Molecular Pharmaceutics Lab., College of Pharmaceutical Sciences, Ritsumeikan University, 1-1-1, Noji-higasi, Kusatsu 525-8577, Shiga, Japan

**Keywords:** liquid–liquid phase separation, intrinsic solubility, melting point, drug-like

## Abstract

The purpose of the present study was to experimentally confirm the thermodynamic correlation between the intrinsic liquid–liquid phase separation (LLPS) concentration ($S_{0}^{LLPS}$) and crystalline solubility ($S_{0}^{c}$) of drug-like molecules. Based on the thermodynamic principles, the crystalline solubility LLPS concentration melting point ($T_{m}$) equation (CLME) was derived ($log_{10}S_{0}^{C}=log_{10}S_{0}^{LLPS}-0.0095\left(T_{m}-310\right)$ for 310 K). The $S_{0}^{LLPS}$ values of 31 drugs were newly measured by simple bulk phase pH-shift or solvent-shift precipitation tests coupled with laser-assisted visual turbidity detection. To ensure the precipitant was not made crystalline at <10 s, the precipitation tests were also performed under the polarized light microscope. The calculated and observed $\log_{10}S_{0}^{C}$ values showed a good correlation (root mean squared error: 0.40 log unit, absolute average error: 0.32 log unit).

## 1. Introduction

Intrinsic aqueous solubility ($S_{0}^{c}$) is one of the most important physicochemical properties of a crystalline drug. Many $S_{0}^{c}$ prediction methods have been reported in the literature [1,2,3]. In most cases, $S_{0}^{c}$ is directly predicted from the chemical structure by empirical statistic approaches [4,5,6]. However, even though this approach has been investigated for more than several decades, $S_{0}^{c}$ prediction is still challenging [7].

An alternative approach would be to separate the solubilization process into the melting and solvation terms based on the theory of thermodynamics (Figure 1).

The general solubility equation (GSE) is one of this kind of stepwise prediction approach. Based on the thermodynamic principles, GSE has been formulated as [8],
(1)$$log_{10}S_{0}^{C}=0.5-log_{10}P_{oct}-0.01\left(T_{m}-T\right)$$
where $T_{m}$ is the melting point, *T* is the temperature, and $P_{oct}$ is the octanol/water partition coefficient. GSE was derived from the thermodynamic principles without parameter fitting. The constant of the GSE (0.5) is attributed to the solubility of a drug in octanol. When a liquid-state drug is completely miscible in octanol, the solubility of the liquid-state drug in octanol is expressed as 0.5 [9].

One merit of this kind of stepwise prediction approach is that it can be used to evaluate the relative contribution of the solvation and crystal lattice energy terms such as $0.5-log_{10}P_{oct}$ and $0.01(T_{m}-T$) in GSE, respectively. Therefore, it is of great help to navigate drug design. In addition, it also helps formulation design because a solubilizing formulation suitable for a drug differs depending on the cause of poor solubility being solvation or crystal lattice energy. Another merit of the stepwise approach is that the intermediate parameters such as $P_{oct}$ and $T_{m}$ can be directly experimentally measured to confirm the prediction accuracy for each process.

Recently, the liquid–liquid phase separation (LLPS) of a drug attracted a lot of attention in drug research [10,11,12,13,14]. Several methods have been reported to measure the intrinsic LLPS concentration ($S_{0}^{LLPS}$) [15]. As explained in the Theory section, LLPS is synonymous with the solvation phenomena shown in Figure 1. Therefore, in theory, $S_{0}^{c}$ can be approximately described by $S_{0}^{LLPS}$ and melting point ($T_{m}$) (see Theory section). However, the accuracy of this approximation has not been experimentally confirmed for drug-like molecules. This information is important for clarifying which energy term is responsible for the inaccuracy of $S_{0}^{c}$ prediction, which may lead to breakthroughs in in silico predictions.

The purpose of the present study was to experimentally confirm the thermodynamic correlation between the intrinsic liquid–liquid phase separation (LLPS) concentration ($S_{0}^{LLPS}$) and crystalline solubility ($S_{0}^{c}$) of drug-like molecules. In this study, the $S_{0}^{LLPS}$ values of 31 drugs were newly measured by a simple precipitation test using laser-assisted visual turbidity detection (LAVTD).

## 2. Theory

The intrinsic solubility ratio of a crystalline drug ($S_{0}^{c}$) and a liquid drug ($S_{0}^{L}$) equals the ideal solubility ratio of the crystal drug ($X_{0}^{c}$) and the liquid drug ($X_{0}^{L}$),
(2)$$\frac{S_{0}^{c}}{S_{0}^{L}}=\frac{X_{0}^{c}}{X_{0}^{L}}$$

The ideal solubility is the mole fraction solubility of a drug in an ideal solution (in the liquid drug). By rearranging this,
(3)$$log_{10}S_{0}^{C}=log_{10}S_{0}^{L}+log_{10}\frac{X_{0}^{c}}{X_{0}^{L}}$$

Assuming that the change in heat capacity upon melting is equal to zero [16], the ideal solubility ratio ($X_{0}^{c}/X_{0}^{L}$) can be estimated as,
(4)$$log_{10}\frac{X_{0}^{c}}{X_{0}^{L}}=-\Delta S_{m}\frac{T_{m}-T}{2.303RT}$$
where $\Delta S_{m}$ is the entropy of melting, and $T_{m}$ is the melting point.

Approximating $S_{0}^{L}$ by the intrinsic liquid–liquid phase separation concentration ($S_{0}^{LLPS}$), $S_{0}^{c}$ is expressed as
(5)$$log_{10}S_{0}^{C}=log_{10}S_{0}^{LLPS}-\Delta S_{m}\frac{T_{m}-T}{2.303RT}$$

This formula is hereinafter referred to as the crystalline solubility LLPS concentration melting point equation (CLME). It should be noted that $S_{0}^{LLPS}$ and $S_{0}^{L}$ do not exactly match because water and a liquid phase drug are mutually miscible to some extent. In addition, LLPS and the glass–liquid phase separation were not distinguished in this study [17].

According to Walden’s rule, $\Delta S_{m}$ = 56.5 J/K∙mol. Therefore, at 298 K (25 °C) and 310 K (37 °C), Equation (5) becomes
(6)$$log_{10}S_{0}^{C}=log_{10}S_{0}^{LLPS}-0.0099\left(T_{m}-298\right)$$
(7)$$log_{10}S_{0}^{C}=log_{10}S_{0}^{LLPS}-0.0095\left(T_{m}-310\right)$$

It should be noted that CLME was derived from the thermodynamic principles without any parameter fitting.

## 3. Materials and Methods

### 3.1. Materials

Sodium hydroxide aqueous solution (8 mol/L) (NaOH), 6 mol/L hydrochloric acid (HCl), sodium chloride (NaCl), sodium dihydrogen phosphate dihydrate (NaH_2_PO_4_ 2H_2_O), *N,N*-dimethylacetamide (DMA), boric acid, methanol, 0.1% trifluoroacetic acid-acetonitrile, (S)-(+)-naproxen, diphenhydramine hydrochloride, haloperidol, ibuprofen, indomethacin, ketoprofen, niflumic-acid, papaverine hydrochloride, (±)-propranolol hydrochloride, quinine, and warfarin sodium were purchased from FUJIFILM Wako Pure Chemical Corporation (Osaka, Japan). 2-Naphthoic acid, acemetacin, bifonazole, bupivacaine hydrochloride, carprofen, chlorpromazine hydrochloride, diclofenac sodium salt, dipyridamole, fenofibrate, flufenamic-acid, flumequine, flurbiprofen, furosemide, glipizide, ketoconazole, ketotifen fumarate, losartan potassium, loxoprofen, mefenamic-acid, meloxicam, phenylbutazone, probenecid, procaine hydrochloride, procaine, propafenone hydrochloride, rebamipide, sulfasalazine, sulindac, thioridazine hydrochloride, and verapamil hydrochloride were purchased from Tokyo Chemical Industry (Tokyo, Japan). Meclofenamic-acid sodium salt and phenytoin sodium were purchased from Sigma-Aldrich (Arklow, Ireland). Benzocaine, lidocaine hydrochloride, and terbinafine hydrochloride were purchased from Combi-Blocks (San Diego, CA, USA). Orphenadrine hydrochloride was purchased from Chem Cruz (Huissen, Netherlands). 0.1% trifluoroacetic acid-distilled water was purchased from Kanto Chemical Co., Inc. (Tokyo, Japan). Pramoxine hydrochloride was purchased from Cayman Chemical (Ann Arbor, MI, USA). Warfarin free acid was prepared by adding 1 N HCl to warfarin sodium dissolved in distilled water. Propafenone free base was prepared by adding 1 N NaOH to propafenone hydrochloride dissolved in distilled water.

### 3.2. Methods

#### 3.2.1. Crystallization Time Measurement

Before the $S_{0}^{LLPS}$ measurements, the crystallization time of 47 drugs was measured by performing the precipitation test under the polarized light microscope (PLM). In the pH-shift precipitation method, an ionizable drug was dissolved in distilled water as a salt form or by adding 1–3 Eq of NaOH (for weak acids) or HCl (for weak bases). A total of 1.0 µL of 1 N HCl (for weak acids) or 1 N NaOH (for weak bases) was dropped onto a glass slide, then the drug solution (9.0 µL) was added and covered with a cover glass. In the solvent-shift precipitation method, an un-dissociable drug was dissolved in *N,N*-dimethylacetamide. Distilled water (19.8 µL) was added to the drug solution (0.2 µL). The drug concentration was set to 30 mM, except for propafenone (20 mM), lidocaine (200 mM), terbinafine (18 mM), fenofibrate (198 mM), and haloperidol (2 mM). For the measurements at 310 K, the temperature was maintained by a glass plate heater (BLAST Inc., Kanagawa, Japan). The precipitants were monitored under a PLM (crossed-Nicols with a sensitive-tint plate) (Olympus CX-43, Olympus Corporation, Tokyo, Japan). The solid state of the precipitant was diagnosed as crystalline when polarization was observed.

#### 3.2.2. $S_{0}^{LLPS}$ Measurement by the Precipitation Tests Coupled with Laser-Assisted Visual Turbidity Detection (LAVTD)

The drugs with a crystallization time > 10 s were selected for the $S_{0}^{LLPS}$ measurements. The $S_{0}^{LLPS}$ value was measured by the bulk phase pH-shift precipitation tests or the solvent-shift precipitation tests coupled with laser-assisted visual turbidity detection (LAVTD) as previously reported [18]. Each drug solution was prepared as described above. For the measurements at 310 K, the drug solution, 1 N NaOH, 1 N HCl, and glass test tubes were pre-heated in a water bath. In the case of ionizable drugs, 1 N NaOH or 1 N HCl (100 µL) was added to the glass test tube, then set to the LAVTD device (Appendix A). The drug solution (900 µL) was then added to the glass test tube and immediately vigorously shaken. In the case of un-dissociable drugs, the drug solution (10 µL) was added to the glass tube, then distilled water (990 µL) was added. Turbidity was visually detected within 10 s with the assistance of a red laser (635 nm). The concentration of the drug solution was changed with 0.001 to 0.1 mM increments to give 3 significant digits. The $S_{0}^{LLPS}$ value was defined as the drug concentration at which the solution started to show turbidity. The $S_{0}^{LLPS}$ measurement was performed in triplicate.

To compare with the literature data, $S_{0}^{LLPS}$ was also measured using the same medium conditions as the reference [19]. Diclofenac sodium was dissolved in methanol. The drug solution was added to the glass test tube (10 µL). Phosphate buffer (990 µL, pH 2.0, PO_4_: 50 mM, NaCl: 128 mM) was added to the glass test tube and immediately vigorously shaken at 298 K. The $S_{0}^{LLPS}$ value was determined as described above.

#### 3.2.3. $S_{0}^{LLPS}$ Measurement by Turbidity Detection Using a UV/VIS Spectrophotometer

Each drug solution was prepared as described above. A 1 N NaOH or 1 N HCl solution (70 µL) was added to the quartz cell and set to a UV/VIS spectrophotometer (UV-1850, Shimadzu Corporation, Kyoto, Japan). A drug solution (630 µL) was rapidly added to the quartz cell and the absorbance was measured at 500 nm within 10 s at 298 K. This wavelength was set to be higher than the absorption wavelength of each drug. The $S_{0}^{LLPS}$ measurement was performed in triplicate.

#### 3.2.4. Intrinsic Solubility Measurement

Crystalline free-form drugs were used for the intrinsic solubility measurement. The intrinsic solubility was measured based on the harmonized protocol as previously reported [20]. Each drug was added to a test solution (10 mL) in a 15 mL tube. The samples were rotated at 40 rpm at 310 K except for procaine (1800 rpm). Before filtration, the sample was allowed to stand still for 1 min. The sample was then filtered (hydrophilic PVDF, 0.22 μm pore size). The first few drops were discarded to avoid filter adsorption [21]. The drug concentration in the filtrate was determined by UV spectroscopy (UV-1850, Shimadzu Corporation, Kyoto, Japan). The residual solid was collected by vacuum filtration and analyzed by differential scanning calorimetry (DSC). The composition of the medium, the amount of the added drug, the incubation time, and the detection wavelength are summarized in Table 1. The achievement of equilibrium was confirmed by time-course measurements up to 48 h.

Procaine showed hydrolysis in alkali conditions [22]. Therefore, the concentration of procaine was determined by HPLC (Shimadzu Prominence LC-20 series; Colum: Zorbax Eclipse Plus C18, 2.1 × 50 mm, particle size: 3.5 μm; mobile phase: acetonitrile/0.1% trifluoroacetic acid (5: 95); flow rate: 0.6 mL/min; temperature: 40 °C; injection volume: 10 μL). The achievement of equilibrium was confirmed by time-course measurements.

#### 3.2.5. Differential Scanning Calorimetry Measurement

The solid form of the residual particles in the intrinsic solubility measurement was determined by differential scanning calorimetry (DSC). The sample was placed in a non-sealed aluminum pan and analyzed by DSC (Shimadzu DSC60 plus, Shimadzu Corporation Kyoto, Japan) under nitrogen gas (50 mL/min). Heat flow was set to 10 °C/min.

#### 3.2.6. Thermodynamic Correlation between the Intrinsic Liquid–Liquid Phase Separation Concentration and Crystalline Solubility

The $S_{0}^{c}$ value was calculated by CLME (Equation (7)). The experimental $T_{m}$ values were obtained from the literature when available (Table 2). The $S_{0}^{c}$ values at 310 K were also obtained from the literature when available. The correlation between the calculated and observed $S_{0}^{c}$ values was evaluated by the average absolute error (AAE), the root mean square error (RMSE), and the coefficient of determination (r^2^). The AAE and RMSE were calculated by
(8)$$AAE=\frac{\sum \left|\log_{10}S_{0,\;calc}^{c}-\log_{10}S_{0,obs}^{c}\right|}{N}$$
(9)$$RMSE=\sqrt{\frac{\sum {\left(\log_{10}S_{0,calc}^{c}-\log_{10}S_{0,obs}^{c}\right)}^{2}}{N}}$$
where the subscript calc and obs indicate the calculated and observed values.

The $S_{0}^{c}$ value was also calculated by GSE using the experimental ${log}_{10}P_{oct}$ values in the literature.

## 4. Results

### 4.1. Crystallization Time

Before the precipitation tests, the crystallization time for each compound was determined under PLM. The results are summarized in Appendix A. Because LAVTD requires 10 s, the drugs that crystallized within 10 s were excluded from the following studies (31 remained out of 47 drugs).

### 4.2. Validation of the Precipitation Tests Coupled with Laser-Assisted Visual Turbidity Detection

In LAVTD, the turbidity of a solution was detected by visual inspection. However, visual detection could cause a measurement error. To validate LAVTD, turbidity measurements were also performed by UV/VIS spectrometry for several drugs (diclofenac, ibuprofen, papaverine, propafenone, and warfarin). The photograph of the LAVTD method and the absorbance vs. concentration profiles measured by the UV/VIS spectrometry are summarized in Figure 2. The $S_{0}^{LLPS}$ value was determined as the concentration intercept value. The $S_{0}^{LLPS}$ of diclofenac, ibuprofen, papaverine, propafenone, and warfarin were determined as 0.25 ± 0.00 mM, 0.64 ± 0.01 mM, 0.83 ± 0.00 mM, 0.38 ± 0.00 mM, and 0.56 ± 0.00 mM, respectively by the UV/VIS method. The $S_{0}^{LLPS}$ values measured by LAVTD were almost identical to those measured by the UV/VIS method (Figure 3). The coefficient of determination was 0.997.

To further validate LAVTD, the $S_{0}^{LLPS}$ value of diclofenac measured by LAVTD was compared with the literature values measured by the UV/VIS method and the fluorescence spectroscopy method [19]. The $S_{0}^{LLPS}$ value measured by the LAVTD methods, the UV/VIS method, and the fluorescence spectroscopy method were 0.20 mM, 0.18 mM, and 0.17 mM, respectively (at 298 K). Therefore, LAVTD yielded the $S_{0}^{LLPS}$ value close to the previously reported values.

### 4.3. Thermodynamic Correlation between the Intrinsic Liquid–Liquid Phase Separation Concentration and Crystalline Solubility

The molecular weight (MW), acid–base dissociation constant ($pK_{a}$), $T_{m}$, $log_{10}P_{oct}$, $log_{10}S_{0}^{LLPS}$, and $log_{10}S_{0}^{C}$ of each drug are summarized in Table 2. Many of the $S_{0}^{LLPS}$ differed from the $S_{0}^{c}$ values more than 10-fold and up to 158-fold. The correlation between the calculated and observed $S_{0}^{C}$ values is shown in Figure 4. The drugs with a melting point below 310 K were excluded from the analysis. In addition, terbinafine was excluded because it was difficult to prepare the crystalline-free base due to its low melting point (314 K). The AAE, RMSE, and r^2^ values are summarized in Table 3.

The correlation between $P_{oct}$ and $S_{0}^{LLPS}$ is shown in Figure 5. When the $P_{oct}$ value increased, the $S_{0}^{LLPS}$ value decreased.
pharmaceutics-14-02560-t002_Table 2Table 2MW, pKa, Tm, Poct, S0LLPS, and S0C of model compounds.DrugsMWpKa 1Tm (K)log10Poctlog10S0LLPS (M)log10S0C (M)Ref.Atazanavir7054.5 (B) 4815.8−4.03−5.83[23,24]Bifonazole3105.7 (B)422 ^2^4.8−4.70 ± 0.00 ^2^−5.42 ± 0.03 ^2^[25,26]Carprofen2744.2 (A) 484 ^2^4.3−3.66 ± 0.00 ^2^−4.80 ± 0.02 ^2^[27]Celecoxib38111.1 (A) 4373.4−3.95−5.50[28,29,30]Chlorpromazine ^3^3199.2 (B) <2985.4−4.70 ± 0.00 ^2^-[27]Cilnidipine493None3875.7−5.33−6.89[31,32,33]Clotrimazole3455.9 (B) 4175.2−4.65−5.80[23,25,34]Clozapine3273.8 (B), 7.5 (B) 4584.1−3.38−4.57[15,25,34]Danazol338None4984.5−4.43−6.21[35,36,37,38]Diclofenac2964.0 (A) 4534.5−3.52 ± 0.00 ^2^−4.96[18,27,39]Diphenhydramine ^3^2559.1 (B)<2983.4−2.94 ± 0.00 ^2^-[27]Dipyridamole5056.4 (B) 4362.2−3.80 ± 0.00 ^2^−4.70[40,41,42]Efavirenz31610.2 (A) 4125.4−4.23−4.59[15,43,44,45]Enzalutamide464None between pH 3–114704.0−4.04−5.20[46,47]Felodipine384<24155.6−4.59−5.61[10,15,25,48]Fenofibrate361None3544.6−4.70 ± 0.00 ^2^−6.08[49,50]Flurbiprofen2444.0 (A) 388 ^2^4.2−3.37 ± 0.00 ^2^−4.15 ± 0.01 ^2^[27]Ibuprofen2064.4 (A) 3494.0−3.12 ± 0.00 ^2^−3.55[27,51]Ketoconazole5313.3 (B), 6.2 (B) 4234.3−3.80 ± 0.00 ^2^−5.31[25,52,53,54]Ketoprofen2544.2 (A) 3683.2−2.76 ± 0.00 ^2^−3.00[25,55,56,57]Ketotifen3096.7 (B)4302.1−3.41 ± 0.00 ^2^−4.28[58,59]Lidocaine2348.0 (B) 3422.4−1.74 ± 0.00 ^2^−1.90[27,60]Loratadine3835.3 (B) 4095.2−4.70−5.38[15,41,61]Losartan4233.2 (A) 4573.5−2.07 ± 0.00 ^2^−3.47[62,63,64,65]Loxoprofen2464.2 (A) 358 ^2^2.3−2.21 ± 0.00 ^2^−2.22 ± 0.00 ^2^[66,67]Meclofenamic-acid2964.1 (A) 5305.9−4.52 ± 0.00 ^2^−6.68[27,68]Miconazole4166.1 (B) 3584.9−4.88−5.62[25,35,69]Orphenadrine ^3^2699.0 (B)<2983.8−3.26 ± 0.00 ^2^-[27]Paclitaxel854None4933.9−4.43−6.38[35,70,71]Papaverine3396.4 (B) 4213.0−3.03 ± 0.00 ^2^−4.35[27,72]Phenylbutazone3084.4 (A) 379 ^2^3.3−3.64 ± 0.00 ^2^−4.49 ± 0.01 ^2^[27]Posaconazole7013.6 (B), 4.6 (B) 4423.8−4.89−6.41[73,74,75]Pramoxine ^3^3307.1 (B)<2983.6−3.09 ± 0.00 ^2^-[27]Procaine2362.3 (B), 9.0 (B)333 ^2^2.1−1.71 ± 0.00 ^2^−2.02 ± 0.01 ^2^[27]Propafenone3419.3 (B) 364 ^2^4.6−3.42 ± 0.00 ^2^−4.62 ± 0.03 ^2^[76,77]Propranolol2599.0 (B)3693.5−2.78 ± 0.00 ^2^−3.07[27,78]Quinine3244.2 (B), 8.6 (B)449 ^2^3.5−2.82 ± 0.00 ^2^−3.26 ± 0.00 ^2^[27]Rebamipide3713.3 (A) 5792.6−3.09 ± 0.00 ^2^−5.29[79,80,81]Ritonavir7212.4 (B) 3913.2−4.58−5.74[15,82,83,84]Sulfasalazine3982.4 (A), 8.0 (A), 10.9 (A) 532 ^2^3.6−4.15 ± 0.00 ^2^−5.92 ± 0.04 ^2^[27]Sulindac3564.1 (A) 460 ^2^3.4−3.70 ± 0.00 ^2^−4.60 ± 0.00 ^2^[27]Telaprevir6800.3 (B), 11.8 (A) 5194.0−3.87−5.17[17,85]Terbinafine ^3^2917.1 (B)3146.2−5.22 ± 0.00 ^2^-[25,86]Thioridazine ^3^3718.9 (B)<2985.3−4.30 ± 0.00 ^2^-[87,88,89]Verapamil ^3^4558.7 (B) <2984.0−4.10 ± 0.00 ^2^-[27]Warfarin3084.9 (A) 436 ^2^3.5−3.20 ± 0.00 ^2^−4.54 ± 0.01 ^2^[27]^1^ A: acid, B: base. ^2^ Measured values in this study. The S0LLPS and S0C values were measured at 310 K. Detailed information on the S0C measurements (DSC curves of residual particles, the pH values of suspension after reaching equilibrium, and calibration curves) are summarized in Appendix A and Figure 5, and Appendix A. ^3^ Excluded from the S0C prediction.
Figure 4Correlation between the calculated and observed S0C values at 310 K by (**A**) Equation (7) (CLME with S0LLPS and Tm) and (**B**) Equation (1) (GSE using Poct). The black dotted line indicates the calculated value equals the observed value.
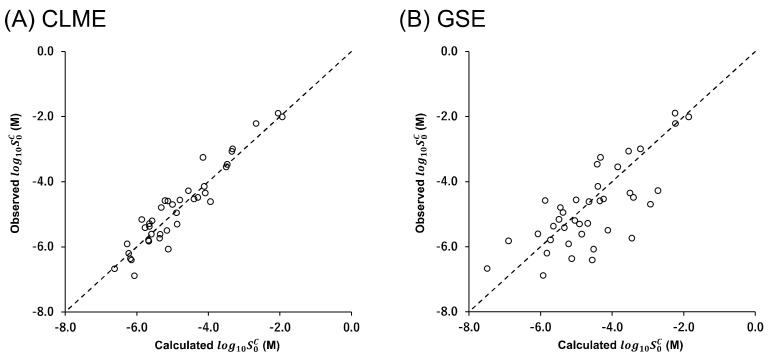

Figure 5Correlation between Poct and S0LLPS. The solid line is log10S0LLPS=0.5−log10Poct.
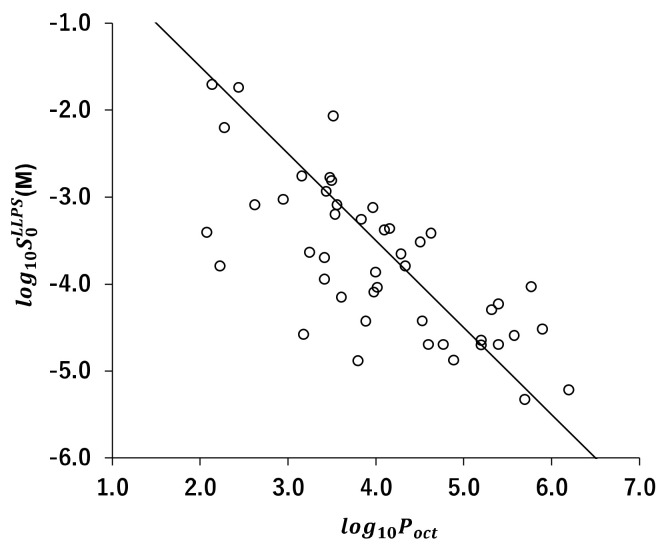

pharmaceutics-14-02560-t003_Table 3Table 3Statistical correlation of CLME and GSE (N = 39).
CLME (with S0LLPS)GSE (Poct)AAE (log unit)0.320.71RMSE (log unit)0.400.91r^2^0.900.56


## 5. Discussion

In this study, we first validated the bulk phase pH-shift and solvent-shift precipitation tests coupled with laser-assisted visual turbidity detection (LAVTD). This method is simple, rapid, robust, and requires minimum instrumental resources (only a red laser pointer). Because LAVTD requires less than 10 s, it enables the $S_{0}^{LLPS}$ measurements of rapidly crystallizing drugs. However, visual detection could cause a measurement error. Therefore, to validate LAVTD, the $S_{0}^{LLPS}$ values were compared with those measured by the UV/VIS spectrophotometric method and the fluorescence spectroscopy method. There was a good agreement between the $S_{0}^{LLPS}$ values measured by these methods.

In the pH-shift LLPS measurements, in the case of ionizable drugs, the pH value was shifted by adding a small volume of 1 N HCl or 1 N NaOH to a drug solution (1:9). This pH-shift procedure can avoid inducing a local high drug concentration that could facilitate drug crystallization. In the case of un-dissociable drugs, a concentration drug solution in DMA was diluted by adding distilled water (the final DMA concentration was 1.0%). In this case, a high drug concentration may be locally formed at the initial stage of dilution. In this study, the LLPS concentration was sought by changing the initial drug concentration rather than stepwise titration with a concentrated drug solution. Stepwise titration changes the concentration of a rich solvent. In addition, during the stepwise titration process, crystallization could be induced before reaching LLPS concentration.

In this study, to shift the pH value of a drug solution, 1 N HCl or 1 N NaOH was used for weak acids and bases, respectively. Therefore, the $S_{0}^{LLPS}$ measurement was conducted at a pH where a drug is undissociated. In other words, LLPS observed in this study is a phenomenon that undissociated non-electrolytes phases separately to form a drug-rich phase (liquid drug phase). The drug-rich phase is a transient state before crystallization. However, when LLPS was formed, a pseudo-equilibrium state was created between the drug-rich phase and the drug molecules molecularly dissolved in water.

In this study, the $S_{0}^{LLPS}$ values of 31 drugs were newly determined. These values were combined with those reported in the literature to be used for the evaluation of CLME (a total of 39 drugs). CLME appropriately described the $S_{0}^{C}$ values. This result encourages that in silico $S_{0}^{c}$ prediction can be improved by dividing the prediction scheme into two processes, $S_{0}^{LLPS}$ prediction and $T_{m}$ prediction. The development of an in silico $S_{0}^{LLPS}$ model is currently under investigation. For drug-like molecules, a correlation was observed between $P_{oct}$ and $S_{0}^{LLPS}$ (Figure 5). Therefore, the same parameters for $P_{oct}$ prediction from chemical structure (hydrogen bonds, molecular volumes, etc.) might also be used to predict $S_{0}^{LLPS}$ [90,91]. A large amount of the $S_{0}^{LLPS}$ data set is required to construct an in silico model. The LAVTD-based method is suitable for high throughput (HTS) measurements. The HTS $S_{0}^{LLPS}$ measurement method is also currently under investigation.

Although the number of drugs used in this study was limited, CLME showed a significantly better correlation than GSE (*p* = 0.0004, comparison of two independent Pearson’s correlation coefficients) (Figure 4). In GSE, $P_{oct}$ is employed to represent the solvation term. In the octanol phase, drug molecules are surrounded by octanol molecules (octanol-drug mixture). On the other hand, in the case of LLPS, in the drug-rich phase, drug molecules are surrounded by themselves (drug-drug mixture). As shown in Figure 1, the solvation process is the same as the partitioning of a drug between the drug-rich phase and the water phase.

There may be three possible ways to further improve the correlation by CLME. First, in this study, the same $\Delta S_{m}$ value (56.5 J/K∙mol) was used for all drugs. However, the $\Delta S_{m}$ values are different for each compound [92]. Second, the activity of the liquid phase (drug-rich phase) saturated with water should be considered. $S_{0}^{LLPS}$ is not exactly the same as $S_{0}^{L}$ because water and a liquid drug phase are mutually miscible to some extent [93,94]. Third, the heat capacity terms should be considered in the ideal solubility ratio calculation [95].

In this study, the $S_{0}^{LLPS}$ values of drugs that crystallized within 10 s were not measured. To measure the $S_{0}^{LLPS}$ value of quickly crystallizing drugs, a crystallization inhibitor such as polymers (e.g., polyvinylpyrrolidone) may be used [96]. However, the $S_{0}^{LLPS}$ value varies depending on the type and concentration of a polymer [97,98]. Therefore, the selection of a polymer for $S_{0}^{LLPS}$ measurements would be important. Alternatively, the results of this study suggest that the $S_{0}^{LLPS}$ value can be calculated from the $S_{0}^{C}$ value and the melting point when the $S_{0}^{LLPS}$ value is difficult to measure.

In conclusion, $S_{0}^{C}$ can be described by CLME with reasonable accuracy. The results of this study are important for a good understanding of drug solubility and shed light on the way to improve in silico $S_{0}^{C}$ prediction. $S_{0}^{LLPS}$ is a drug intrinsic parameter that determines the maximum achievable concentration of molecularly dissolved drugs in aqueous media. The $S_{0}^{LLPS}$ measurement using LAVTD is simple and easy. It would be especially useful for highly lipophilic drugs for which $S_{0}^{C}$ and $P_{oct}$ measurements are often difficult. Therefore, similar to the other drug intrinsic parameters such as p*K_a_* and $P_{oct}$, $S_{0}^{LLPS}$ should be routinely measured in drug discovery.

## Figures and Tables

**Figure 1 pharmaceutics-14-02560-f001:**
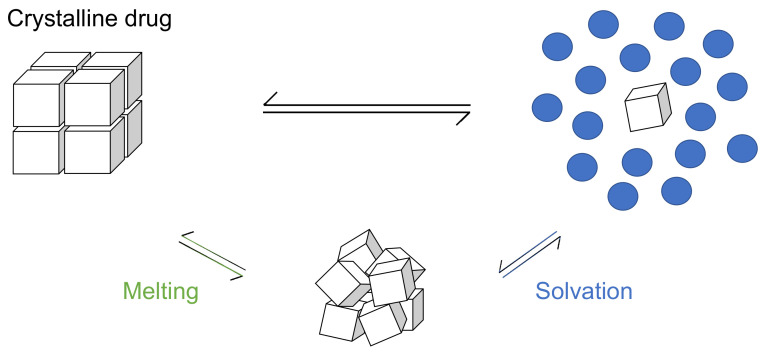
Thermodynamic scheme of crystalline drug solubilization.

**Figure 2 pharmaceutics-14-02560-f002:**
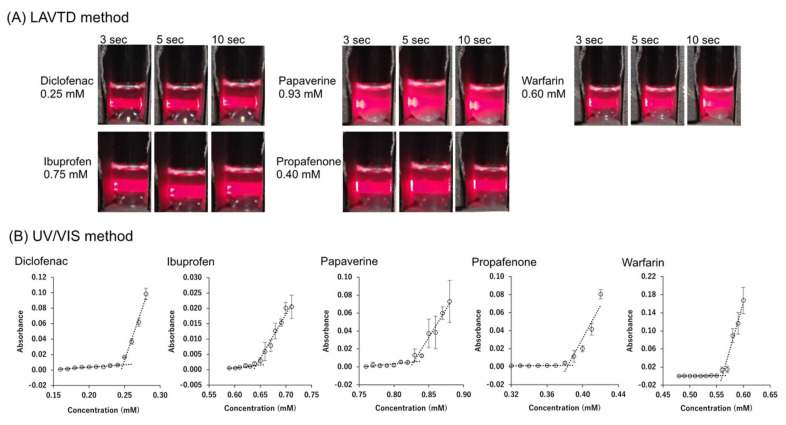
(**A**) The photograph of LAVTD method at 298 K. (**B**) Concentration vs. absorbance profiles in $S_{0}^{LLPS}$ measurement by turbidity detection by UV/VIS spectroscopy at 298 K. The $S_{0}^{LLPS}$ value was determined as the concentration intercept value.

**Figure 3 pharmaceutics-14-02560-f003:**
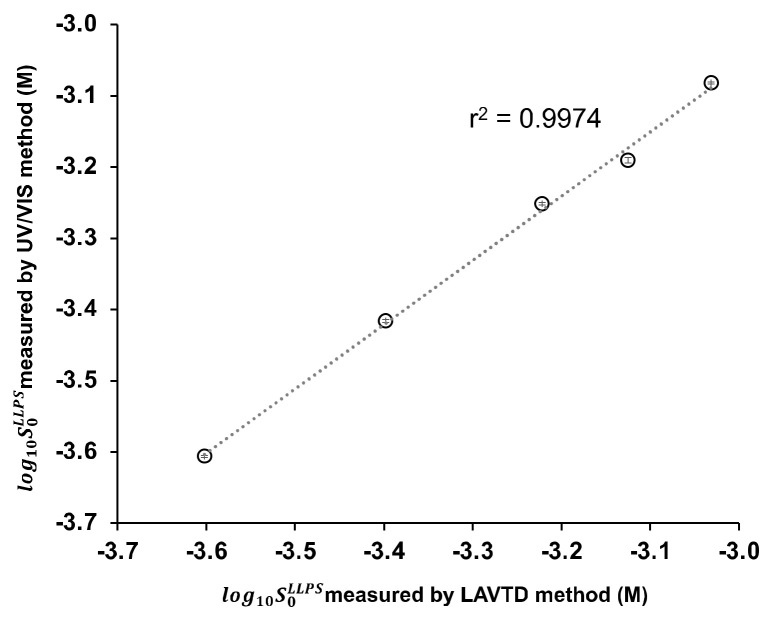
Comparison of the $S_{0}^{LLPS}$ values measured with the UV/VIS method and the LAVTD method (mean ± S.D., n = 3) (298 K).

**Table 1 pharmaceutics-14-02560-t001:** Experimental conditions of intrinsic solubility measurement.

Drug	Medium	Amount of Drug (mg)	Incubation Time (h)	Wavelength (nm)
Bifonazole	pH 9.0 borate buffer ^1^	30	72	255
Carprofen	0.1 N HCl	30	48	300
Flurbiprofen	0.1 N HCl	30	48	248
Loxoprofen	0.1 N HCl	50	48	220
Phenylbutazone	0.1 N HCl	30	48	264
Procaine	0.01 N NaOH	500	1	280
Propafenone	0.01 N NaOH	30	48	305
Quinine	0.01 N NaOH	100	48	350
Sulfasalazine	0.1 N HCl	30	48	369
Sulindac	0.1 N HCl	50	48	331
Warfarin	0.1 N HCl	30	48	275

^1^ The concentration of boric acid was adjusted to 50 mM.

## Data Availability

The data presented in this study are available on request from the corresponding author.

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
