# Peer review of "Thermodynamic Correlation between Liquid–Liquid Phase Separation and Crystalline Solubility of Drug-Like Molecules"

_pharmaceutics, 2022, doi:10.3390/pharmaceutics14122560_

Round 1
Reviewer 1 Report
I do not recommend this manuscript for publication because the discussion of the GSE is unfair and inappropriate
The authors should consider the following comparison between the GSE and SOLLPS before considering resubmitting their manuscript:
In the manuscript the authors indicate that SOLLIPS was evaluated (or fitted) to a small data set and does not use any kind of test set, whereas the GSE has been evaluated on over 1000 compounds encompassing a diverse range of chemical structure. With only 31 drugs, SOLLPS cannot cover much structural diversity.
The GSE is easy to use with (if using commercial databases) requires no experimentation, whereas SOLLPS requires a cumbersome procedure. It would be easier for a person to measure the solubility than to perform the procedure described in the manuscript. Also, the solubility measurement is theoretically at equilibrium whereas the precipitation is not.
Since the GSE can be performed on non-existing compounds, it can be used in drug design. SOLLPS can’t.
Since the GSE is defined as using logP. Therefore, SOLLPS cannot be regarded an improvement of the GSE. It is a totally different model.
The GSE uses 0.01 (MP-25) it does not use 0.0099(Tm-298). Why does the author repeatedly print 0.0099? Especially when they state that the logP is approximate.
The GSE estimates the solubility of undissociated non-electrolytes. This may not be the case in the microenvironment of the precipitate that is forming in the proposed procedure.
Reviewer 2 Report
Pharmaceutics-1963079
Application of liquid-liquid phase separation concentration to Yalkowsky’s general solubility equation
Comments and Questions:
The use of intrinsic liquid-liquid phase separation concentration instead of the octanol/water partition coefficient was verified experimentally by 31 drugs that it can improve the prediction accuracy of the general solubility equation.
This paper should be accepted after some major corrections:
1. Please say a little bit more about the “0.5” in Eq (1).
2. Why LLPS is more directly related to solvation phenomena then octanol/water partition?
3. Do all drugs have intrinsic liquid-liquid phase separation concentration in water or aqueous media?
4. Could Yalkowsky’s general solubility equation be applied to other solvent systems? Or it is an aqueous solubility equation?
5. Photo images of the turbidity change by the LAVTD method should be included for some time points for the 5 drugs in Figure 2 as well.
6. Tangent lines should be included in Figure 2 to show how to obtained the intrinsic liquid-liquid phase separation concentration values.
7. Optical macrographs should be provided to demonstrate how to perform crystallization time measurement properly.
8. Were the non-sealed DSC aluminum pans punched with a hole? Did some of the drugs sublime or decompose during heating? Please include the DSC scans of all 31 drugs and their melting endotherms in the supporting information.
9. Can a hot-stage optical microscopy or melting point apparatus be used instead of DSC?
10. Could the 31 drugs be categorized into groups based on their intrinsic liquid-liquid phase separation concentration values? Are the values related to the molecular structures or functional groups? Please show the molecular structure of those 31 drugs in the Supporting Information.
11. Why the partition coefficient and intrinsic liquid-liquid phase separation concentration have an opposite trend in Fig. 5? What is the physical meaning of that?
12. Can the instrinsic aqueous solubility be measured directly, for example, by the gravimetric titration (initial solvent screening) method visually with the help of a laser light pen and UV-vis instead but without the measurements of the liquid-liquid phase separation concentration and melting point? Please discuss and compare the methods (Please see: Lee, T.; Kuo, C. S.; Chen, Y. H., “Solubility, polymorphism, crystallinity, and crystal habit of acetaminophen and ibuprofen by initial solvent screening,” Pharm. Tech. 2006, 30(10), 72-92).
13. Are there any other direct ways to measure the intrinsic aquoeus solubility of drugs which are competitive to your method?
Reviewer 3 Report
The article „Application of liquid-liquid phase separation concentration to Yalkowsky’s general solubility equation” submitted to the Physico Chemical Profiling Pharmaceutics: Solubility and Permeability Special Issue of Pharmaceutics is an interesting and well-executed piece of work.
The aim of the Authors was to confirm if using the liquid-liquid phase separation concentration instead of the widely used octanol/water partition coefficient in the general solubility equation can improve its effectiveness. This approach was validated by using a dataset of experimentally determined liquid-liquid phase separation concentration values and indeed the tested approach offers a greater accuracy of solubility prediction compared to the standard equation. The concept of introducing a different variable to the general solubility equation is quite simple yet effective which is the advantage of the presented approach. The Authors have conducted a number of experiments and presented a simple way of determining the liquid-liquid phase separation concentration which was validated. When discussing their findings the Authors are aware of the limitations of the proposed approach and indicate areas for its future development. Overall, the scientific value of the paper is high, the computational approach is simple and reasonable, the experiments are well-designed, the paper is easy to read and the topic is interesting for the scientific community.
The paper would be a valuable addition to the Physico Chemical Profiling Pharmaceutics: Solubility and Permeability Special Issue of Pharmaceutics after addressing the following issues.
The Authors describe some drugs as having a melting point lower than 298 K. However Diphenhydramine (CAS: 58-73-1) for example is supposed to have a melting point of about 170 degrees Celsius (see: https://pubchem.ncbi.nlm.nih.gov/compound/Diphenhydramine#section=Melting-Point). Also, shouldn’t such a low melting point of these drugs interfere with the crystallization time measurements and the liquid-liquid phase separation concentration measurements? Please check this.
Please provide CAS numbers for the active pharmaceutical ingredients in point 3.1.
Please indicate in Table 2 at which temperature the intrinsic solubility was measured.
The intrinsic solubility was determined via a spectrophotometric method. If calibration curves were used then adding their details (linear equation, determination coefficient) in supplementary materials might be useful for other researchers.
Round 2
Reviewer 1 Report
The following are my suggestions:
Introduction
1. Change reference 8 to Jain and Yalkowsky (2001) “Estimation of the aqueous solubility 1: Application to organic nonelectrolytes”. J. Pharm. Sci 90,(2) 234-252. The Valvani paper is partly empirical and has a different intercept.
2. The derivation followed that of Jain and Yalkowsky, but it is presented as though it is their original idea.
Experimental
1. Did the authors use pure water or water saturated with octanol ?
2. The Experiments are kinetic and likely do not represent true equilibrium. There is the possibility of slow equilibration or supersaturation.
3. Note that many of the SLLPS values differ from the SC values by more than a factor of ten
And the values of five of the compounds differ by more than a factor of 50
4. Also, 10 s is too short a time for equilibrium solubility to be attained.
If the authors want to compare their data to the GSE they must use equilibrium solubility data
Reviewer 2 Report
None.
